# Beneficial Effect of Curved Dilator System for Femoral Tunnel Creation in Preventing Femoral Tunnel Widening after Anterior Cruciate Ligament Reconstruction

**DOI:** 10.3390/medicina59081437

**Published:** 2023-08-08

**Authors:** O-Sung Lee, Joong Il Kim, Seok Hyeon Han, Joon Kyu Lee

**Affiliations:** 1Department of Orthopedic Surgery, Eulji University School of Medicine, Uijeongbu-si 11759, Republic of Korea; xixzeus@naver.com; 2Department of Orthopaedic Surgery, Hallym University Kangnam Sacred Heart Hospital, Seoul 07741, Republic of Korea; jungil@hanmail.net; 3Department of Orthopaedic Surgery, Konkuk University Medical Center, Seoul 05030, Republic of Korea; shh3892@naver.com; 4Department of Orthopaedic Surgery, Konkuk University Medical Center, Research Institute of Medical Science, Konkuk University School of Medicine, Seoul 05030, Republic of Korea

**Keywords:** bone tunnel widening, bone tunnel enlargement, anterior cruciate ligament, anterior cruciate ligament reconstruction

## Abstract

*Backgrounds and objectives:* A prevalent concern in anterior cruciate ligament (ACL) reconstruction is postoperative tunnel widening. We hypothesized that employing a curved dilator system (CDS) for femoral tunnel creation can reduce this widening after ACL reconstruction compared to the use of a conventional rigid reamer. *Materials and Methods:* A retrospective study was conducted involving 56 patients who underwent primary ACL reconstruction between January 2012 and July 2013. The patients were categorized into two groups: the reamer group (*n* = 28) and CDS group (*n* = 28). All participants were followed up for a minimum of 2 years. Clinical assessment included the Lachman test and pivot-shift test, and the Lysholm score and subjective International Knee Documentation Committee scores. Radiographic evaluation covered the tunnel widening rate, represented as the ratio of the tunnel diameter 2 years after surgery to the tunnel diameter immediately after surgery, and the ratio (A/B) of femoral tunnel (A) to tibial tunnel (B) diameters at respective time points. *Results:* No significant disparities were found between the two groups in terms of clinical outcomes. However, the reamer group exhibited a greater femoral tunnel widening rate compared to the CDS group (reamer group vs. CDS group: 142.7 ± 22.0% vs. 128.0 ± 19.0% on the anteroposterior (AP) radiograph and 140.8 ± 14.2% vs. 122.9 ± 13.4% on the lateral radiograph; all *p <* 0.05). Two years post-operation, the A/B ratio rose in the reamer group (0.96 ± 0.05→1.00 ± 0.05 on the AP radiograph and 0.94 ± 0.03→1.00 ± 0.0.04 on the lateral radiograph; all *p* < 0.05), while it decreased in the CDS group (0.99 ± 0.02→0.96 ± 0.05 on the AP radiograph and 0.97 ± 0.03→0.93 ± 0.06 on the lateral radiograph; all *p* < 0.05). *Conclusion:* The use of CDS for femoral tunnel creation in primary ACL reconstruction provides a potential advantage by limiting tunnel widening compared to the conventional rigid-reamer approach.

## 1. Introduction

The issue of tunnel widening following anterior cruciate ligament (ACL) reconstruction is a well-documented complication, which could potentially lead to surgical failure. [1,2,3,4]. It not only jeopardizes the graft success but also requires staged management in revision ACL reconstruction [5,6]. While the precise cause of tunnel widening remains elusive, contributing factors have been suggested to be biological, biomechanical, and mechanical [7,8,9,10]. Biological factors include healing at the graft–tunnel interface and increased cytokine levels leading to inflammatory responses, infection, and cell apoptosis. Biomechanical factors include foreign body reactions and necrosis due to heat during drilling for tunnel creation. And, mechanical factors include the graft fixation method, the graft motion within the tunnel, the position of the tunnel, and accelerated rehabilitation. 

Among these mechanical factors, different outcomes related to tunnel widening have been reported with varying graft fixation methods, graft types, tunnel locations, and femoral tunnel creation methods [11,12,13,14,15,16]. However, the literature offers scarce comparative data on tunnel widening after using a dilator versus a conventional reamer in ACL reconstruction. We introduce the curved dilator system (CDS) in order to overcome the weakness of the anteromedial (AM) portal technique with a conventional reamer, which may hinder the attainment of an anatomical position for the femoral tunnel [17]. By using CDS for femoral tunnel creation, we theorize that it might condense the inner wall of the tunnel and hence mitigate tunnel widening. Consequently, we posit that utilizing the CDS in ACL reconstruction could potentially reduce both mechanical and biomechanical tunnel widening in comparison to the conventional rigid reamer. Therefore, this study aims to examine tunnel enlargement following ACL reconstruction conducted using the CDS versus the conventional reamer.

## 2. Materials and Methods

We examined individuals who underwent initial ACL reconstruction from January 2012 to July 2013. The inclusion criteria were primary ACL reconstruction using the allogenous tibialis tendon. Only those patients who underwent both clinical and radiological assessments at a 2-year postoperative follow-up were incorporated in the study. The exclusion criteria were as follows: (1) previous intra-articular ligament reconstruction, (2) multiple-ligament injury, and (3) previous osseous procedures. A concomitant meniscal injury was not an exclusion criterion. The institutional review board granted approval for this research.

### 2.1. Surgical Procedure

Surgical procedures exhibited no variations, with the exception of the technique employed for femoral tunnel creation between the groups. The portals utilized included anterolateral, AM, and accessory anteromedial (AAM). Notably, the AM portal was positioned slightly more proximal than the standard approach. In contrast, the AAM portal was created 10–15 mm medial to the medial border of the patellar tendon, representing a notably lateral shift from the usual AAM portal placement and situated distally at the level immediately proximal to the lateral meniscal superior surface [17]. 

In the reamer group, the procedure involved inserting a guide pin through a drill guide, targeting the center of the ACL footprint. A conventional rigid reamer with the same diameter as the graft was utilized to create the femoral tunnel. On the other hand, in the CDS group, a 4.5 mm-diameter curved guide trocar with a sharp end was introduced at the anatomical footprint of the ACL. Subsequently, another 4.5 mm-diameter curved guide trocar with a sharp end was carefully inserted at a marked point while the knee was flexed to slightly over 90°, which was the optimal angle to prevent damage to the medial condyle cartilage during trocar passage. The trocar was then gently hammered until it completely penetrated the far cortex of the lateral condyle. Afterward, the trocar was removed, and the tunnel was gradually widened in incremental steps, increasing by 1 mm in diameter at a time to match the graft’s diameter.

For creating the tibial tunnel, the entry point was positioned at the level of the tibial tubercle, 2–3 cm medial to the tubercle, just above the attachment site of the pes anserinus. A guide pin was inserted at a 55° angle to the tibial plateau, guided by a tibial drill, aiming at the center of the ACL footprint. The tibial tunnel was then drilled along the guide pin using a standard reamer with a diameter that corresponded to that of the graft.

All patients in both groups conducted the same standardized home-based rehabilitation protocol. Routine follow ups were conducted at 2 weeks, 6 weeks, 3 months, 6 months, 1 year, and annually thereafter to ensure appropriate rehabilitation for each phase.

### 2.2. Clinical and Radiological Assessment

Clinical and radiological assessments were conducted prior to the surgery, at 3 months, 12 months after the operation, and annually thereafter. The clinical and radiological outcomes achieved 2 years after the operation were utilized for comparing the groups.

The assessment of knee joint stability included the Lachman test and pivot-shift test. The Lachman test measured anterior translation compared to the uninvolved side and was graded from 0 to 3 (0, <2 mm; 1, 2–5 mm; 2, 5–10 mm; 3, >10 mm). The pivot-shift test was evaluated against the uninvolved side and graded from 0 to 3 (0, same as uninvolved side; 1, gentle gliding; 2, clunk; 3, locking). Clinical status was evaluated 1 day before the surgery and annually thereafter using the Lysholm score and subjective International Knee Documentation Committee (IKDC) scores. A Lysholm score consists of eight items, including limping, locking, pain, stair climbing, the use of supports, instability, swelling, and squatting. Subjective IKDC scores include 18 items covering the domains of symptoms (7 items), sports activities (10 items), and function (1 item). The IKDC-SKF ranges from 0 to 100; higher scores indicate higher levels of function and fewer symptoms. 

For radiological assessment, standardized anteroposterior (AP) and lateral knee radiographs were taken to evaluate tunnel widening. The diameter of the femoral and tibial tunnels was measured based on the method by L’Insalata et al. [2]. This involved measuring the length between the inner borders of the sclerotic margins perpendicular to the tunnel axis on the AP and lateral radiographs (Figure 1). Radiographs obtained immediately after the surgery and 2 years postoperatively were used for evaluation.

All radiological parameters were measured twice, with a 2-week interval and using a Picture Archiving and Communication System, by two orthopedic surgeons who were not part of the surgical team. To assess tunnel widening, a tunnel widening rate was calculated, representing the ratio of the tunnel diameter after 2 years of surgery to the tunnel diameter immediately after surgery. Additionally, the ratio (A/B) of the diameter of the femoral tunnel (A) and tibial tunnel (B) was calculated at each point to account for slight differences in the graft diameter among cases.

### 2.3. Statistical Analysis

Continuous variables were expressed as means with standard deviations, and their comparison was conducted using Student’s *t*-test. Categorical variables, on the other hand, were analyzed using the Pearson chi-square test. The inter- and intraobserver reliabilities of radiological measurements were determined to be acceptable based on the calculated intra-class correlation coefficients. Two orthopedic surgeons working in the knee division of the orthopedic department performed two measurements, at 1-week intervals, and the inter- and intra-observer reliability for the radiologic measurements were 0.89 (range, 0.87–0.97) and 0.91 (range, 0.85–0.96), respectively. A post hoc power analysis was conducted using G power calculator 3.1.9.2 to ensure that the sample size for this retrospective study was sufficient. All statistical analyses were carried out using SPSS software (ver. 25.0.0), and statistical significance was set at a *p*-value of less than 0.05.

## 3. Results

Among a total of 60 consecutive patients who underwent primary ACL reconstruction, 4 patients were excluded due to not meeting the selection criteria or inadequate follow up. These exclusions were based on the following reasons: less than 2 years of follow up for two patients, one patient experienced postoperative complications related to an infection, and one patient had missing clinical data or radiographs.

Ultimately, 56 patients with a minimum 2-year follow up were included in the study and divided into two groups based on the surgical instrument used for femoral tunneling: the reamer group (standard rigid reamer, *n* = 28), which involved a standard rigid reamer, and the CDS group (curved dilator system, *n* = 28), which involved a curved dilator system. Patients’ demographic data, including age, sex, and body mass index, were retrieved from medical records and are presented in Table 1. Demographic characteristics showed no significant differences between the two groups.

### Clinical and Radiological Results

Clinical evaluations of knee joint stability, including the Lachman test and pivot-shift test, revealed no significant differences between the two groups. Similarly, no significant differences were found in the Lysholm score and subjective IKDC score between groups (Table 2).

The femoral tunnel widening rate on the AP radiograph was greater in the reamer group than in the CDS group (reamer group, 142.7 ± 22.0 vs. CDS group, 128.0 ± 19.0; *p =* 0.014). The femoral tunnel widening rate on the lateral radiograph was also larger in the reamer group than in the CDS group (reamer group, 140.8 ± 14.2 vs. CDS group, 122.9 ± 13.4; *p =* 0.001). However, the tibial tunnel widening rate on both the AP and lateral radiographs showed no statistically significant differences in both groups (Table 3).

Regarding the ratio (A/B) of the diameters of the femoral tunnel (A) and tibial tunnel (B), the ratio increased in the reamer group from immediately after surgery to the last follow up (AP radiograph: 0.96 ± 0.05 to 1.00 ± 0.05, *p* < 0.001; lateral radiograph: 0.94 ± 0.03 to 1.00 ± 0.04, *p* < 0.001). Conversely, in the CDS group, the ratio decreased (AP radiograph: 0.99 ± 0.02 to 0.96 ± 0.05, *p* = 0.001; lateral radiograph: 0.97 ± 0.03 to 0.93 ± 0.06, *p* = 0.020) (Table 4).

## 4. Discussion

The main findings of our investigation are as follows: (1) Femoral tunnel widening was significantly larger in the reamer group than in the CDS group 2 years following primary ACL reconstruction. (2) The diameter ratio of the femoral to tibial tunnel decreased in the CDS group but increased in the reamer group 2 years after surgery. These findings suggest that using a curved dilator system (CDS) to create the femoral tunnel in ACL reconstruction could be advantageous in preventing femoral tunnel widening, compared to the conventional rigid reamer.

Tunnel enlargement is a widely recognized complication after ACL surgery since the early 1990s [1,2,18]. The prevalence of tunnel widening has been documented, ranging in variability from 20% to 100% from several studies [11,13,14,19,20,21]. This complication is concerning as it can lead to ACL reconstruction failure due to unsuitable conditions for graft fixation. Moreover, it may necessitate complex, staged management for revision ACL surgery, including initial bone grafting procedures to fill the widened tunnels [10,22,23]. This complication is not only related to surgical techniques but also to the type of fixation devices, graft selection, and demographic factors [12,15,16,19,24,25]. The relation between tunnel widening and clinical outcomes remains uncertain. Some studies have hypothesized that tunnel widening potentially serves as an early finding of graft failure. However, many recent studies could not find any relationship between tunnel enlargement and the clinical results after surgery, and only a few have demonstrated the impact on the clinical outcome of tunnel widening [15,16,21,25,26,27].

While the precise causes of tunnel enlargement are very partially understood, it is generally agreed upon that multiple factors, including biological, biomechanical, and mechanical, contribute to its occurrence. Several studies reported which factors affect tunnel widening, such as the types of grafts, the type of fixation device, the position and size of the tunnel holes, the micro-motion of the graft in the tunnel, and altered cytokine levels in the synovial joint fluid [28,29,30,31,32,33]. Biological factors such as graft healing at the tunnel surface, inflammatory reactions, abnormal activity of the osteoclast, and infection are implicated in tunnel widening after ACL reconstruction. A biomechanical factor is heat necrosis from drilling and foreign bodies. A certain amount of heat by drilling may induce the necrosis of the bone and secondary inflammation. And, several studies have reported that various types of allogenic grafts may raise the risk for tunnel enlargement with the foreign-body immune response [1,20,34,35]. Several studies reported the graft type, cyclic loading of the knee joint, and motion of the graft within the tunnel as mechanical factors after ACL reconstruction [26,29,36]. The longitudinal motion of the graft along the tunnel, known as the bungee effect, and the transverse motion of the graft perpendicular to the axis of the tunnel known as the windshield-wiper effect may lead to subsequent bone resorption. 

Recently, surgeons suggested some possible treatments in order to mitigate the tunnel widening. A preclinical study showed that the utilization of mesenchymal stem cells seeded in a collagen type-I scaffold enhanced the biological healing of ACL grafts in a rabbit model [37]. Additionally, Robinson J. et al. reported that a poly-L-lactic acid-hydroxyapatite-blended bio-absorbable screw reduced the tunnel widening after ACL surgery using the hamstring tendon [19]. Yamazaki S. et al. demonstrated in their study that the utilization of transforming growth factor-beta 1 substantially enhanced the production of collagen fibers linking the tendon graft to bone in dogs [38]. And, Hashimoto Y. et al. reported that recombinant BMP-2 treatment resulted in the successful regeneration of the tendon–bone junction in a rabbit model [5]. Additionally, hyperbaric oxygen significantly promoted the incorporation of bone to tendon, and rose the tensile strength of the graft by enhancing the amount of trabecular bone around the graft [17].

Our hypothesis is that continuous dilators for the creation of the femoral tunnel might reduce the tunnel widening compared to conventional drilling with a rigid reamer by creating a more condense tunnel wall. Previous studies have reported varying results regarding tunnel widening according to the type of reamer used for tunnel drilling [13,17]. Justin R. Knight et al. observed significantly greater tibial tunnel deformation when using the acorn reamer compared to the mono-fluted reamer [17]. However, Rainer Siebold et al. did not find a significant reduction in postoperative tunnel widening after ACL reconstruction using compaction drilling with a stepped router [17]. In our investigation, we sought to determine if the creation of the femoral tunnel using CDS could effectively decrease femoral bone tunnel widening following ACL reconstruction, and our hypothesis was strongly supported.

Despite these findings, our study has some limitations. Firstly, it was retrospective and lacked randomization, potentially leading to selection bias. However, we carefully compared the two groups, ensuring no significant differences in basic demographic data, thereby minimizing confounding factors. Secondly, the evaluation period was limited to 2 years after surgery, and further investigation is required to assess whether changes in tunnel diameter affect long-term clinical outcomes after ACL reconstruction. Thirdly, we did not employ three-dimensional imaging modalities like CT or MRI to evaluate tunnel diameters. Nonetheless, the method used in our study, as established by L’Insalata et al., has been widely accepted as a standard approach for measuring tunnel diameter after ACL reconstruction in numerous published studies.

## 5. Conclusions

The femoral tunnel widening rate was larger in the reamer group than in the CDS group on both the AP and lateral radiographs, and the ratio (A/B) of the diameter of the femoral tunnel (A) and tibial tunnel (B) decreased in the CDS group but increased in the reamer group 2 years after ACL reconstruction. These results suggest that using CDS in primary ACL reconstruction may help prevent femoral tunnel widening, compared to the conventional rigid reamer. Nevertheless, there were no notable disparities in clinical results among the two groups during the 2-year follow up. Additional investigations with extended follow-up durations are warranted to comprehensively understand the long-term implications of our findings.

## Figures and Tables

**Figure 1 medicina-59-01437-f001:**
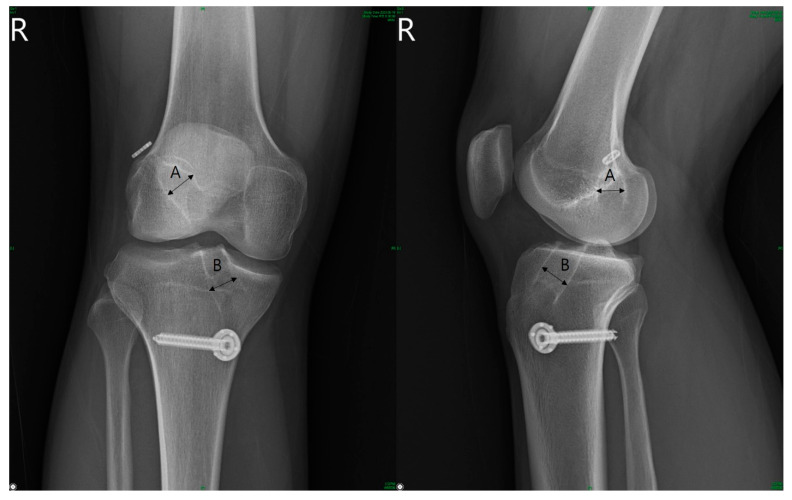
Tunnel diameter was determined as the widest distance between the sclerotic margins of the tunnel perpendicular to the axis of the tunnel based on the method by L’Insalata et al. The diameters were measured from the AP and lateral radiograph of the operated knee at 2 years post-operation.

**Table 1 medicina-59-01437-t001:** Demographic data of patients in the reamer and CDS groups.

	Reamer	CDS	*p*-Value
No. of knees	28	28	
Age, years	30.2 ± 11.1 (17~54)	28.6 ± 12.7 (17~57)	0.537 †
Male/Female	24/4	25/3	0.521 *
Right/Left	15/13	16/13	0.900 *
BMI, kg/m^2^	24.1 ± 2.6	25.2 ± 3.4	0.215 †

BMI, body mass index; CDS, curved dilator system. The values are presented as the mean ± standard deviation with the range in parentheses. † Derived using Student’s *t*-test. * Derived using the Pearson chi-square test.

**Table 2 medicina-59-01437-t002:** Comparison of the clinical outcomes between the reamer and CDS groups 2 years after surgery.

	Reamer (*n* = 28)	CDS (*n* = 28)	*p*-Value
Lachman test (Grade, 0:1:2:3)	4:19:4:1	6:17:3:2	0.804 *
Pivot-shift test (Grade, 0:1:2:3)	9:15:3:1	7:16:4:1	0.833 *
Lysholm score	89.4 ± 10.9	91.8 ± 8.6	0.392 †
IKDC scores	80.2 ± 8.8	77.8 ± 10.1	0.427 †

CDS, curved dilator system; IKDC, International Knee Documentation Committee. The values are presented as the mean ± standard deviation with the range in parentheses. † Derived using Student’s *t*-test. * Derived using the Pearson chi-square test.

**Table 3 medicina-59-01437-t003:** Comparison of tunnel widening rate (percent) between the reamer and CDS groups using the plain radiograph 2 years after surgery.

		Reamer (%)	CDS (%)	*p*-Value
Femoral tunnel	AP	142.7 ± 22.0	128.0 ± 19.0	0.014
	Lateral	140.8 ± 14.2	122.9 ± 13.4	0.001
Tibial tunnel	AP	133.9 ± 16.4	132.7 ± 19.7	0.821
	Lateral	131.7 ± 11.0	127.5 ± 13.0	0.246

CDS, curved dilator system. The tunnel widening rate from immediately to 2 years after surgery was calculated. The values are presented as the mean ± standard deviation. The statistical significance was set at *p* < 0.05 and derived using Student’s *t*-test.

**Table 4 medicina-59-01437-t004:** Ratio (A/B) of the femoral tunnel (A) and tibia tunnel (B) immediately and 2 years after surgery.

		Postoperative, Immediate	Postoperative, 2 Years	*p*-Value
Reamer	AP	0.96 ± 0.05	1.00 ± 0.05	<0.001
	Lateral	0.94 ± 0.03	1.00 ± 0.04	<0.001
CDS	AP	0.99 ± 0.02	0.96 ± 0.05	0.001
	Lateral	0.97 ± 0.03	0.93 ± 0.06	0.020

CDS, curved dilator system. The values are presented as the mean ± standard deviation. The statistical significance was set at *p* < 0.05 and derived using Student’s *t*-test.

## Data Availability

The data presented in this study are available on request from the corresponding author. The data are not publicly available due to privacy.

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
