# Peer review of "Beneficial Effect of Curved Dilator System for Femoral Tunnel Creation in Preventing Femoral Tunnel Widening after Anterior Cruciate Ligament Reconstruction"

_medicina, 2023, doi:10.3390/medicina59081437_

Round 1

Reviewer 1 Report

The authors compare the results for ACL reconstruction using two techniques for femoral tunnel creation: reaming through antero-medial approach versus curved dilator system. Actually, they are comparing the femoral tunnel widening at 2 years from surgery. Since widening of the femoral tunnel is a well-known and potentially dangerous complication of ACL reconstruction, this comparison might be of practical use for the attending orthopedic surgeons Even though CDS system use is not the only technical option to avoid the tunnel widening, their results demonstrate it is effective. The study is well conducted and the results are very nicely presented.

There are some minor issues

Fig. 1 – Are these pictures from the same patient? When were they acquired? Immediately after surgery or 2 years later? It would be useful to mention these details.

References are not numbered in order of appearance in the text and are numbered twice in the reference list

Author Response

We have changed our article as you suggested. We have highlighted the changes of our manuscript within the document by using red text and have added a detailed description on the areas that you have suggested.

Fig. 1 – Are these pictures from the same patient? When were they acquired? Immediately after surgery or 2 years later? It would be useful to mention these details.

: Thank you for your advice. We added detailed mention as you recommended

References are not numbered in order of appearance in the text and are numbered twice in the reference list

: We numbered references in order of appearance, and removed double numbering.

Reviewer 2 Report

Exclusion criteria should be added. Inclusion criteria should be expanded and more detailed information should be given.

In the material method section, a descriptive explanation of Lysholm score and subjective International Knee Documentation Committee (IKDC) scores can be given with a few sentences.

Information about inter-rater and intrarater reliability is not provided in the measurements made. This can be measured using statistical methods such as Cohen's kappa coefficient, intraclass correlation coefficient (ICC), or Fleiss' kappa.

Author Response

We have changed our article as you suggested. We have highlighted the changes of our manuscript within the document by using red text and have added a detailed description on the areas that you have suggested.

Exclusion criteria should be added. Inclusion criteria should be expanded and more detailed information should be given.

: We added exclusion and inclusion criteria. Thank you.

In the material method section, a descriptive explanation of Lysholm score and subjective International Knee Documentation Committee (IKDC) scores can be given with a few sentences.

: We added the explanation as you recommended.

Information about inter-rater and intrarater reliability is not provided in the measurements made. This can be measured using statistical methods such as Cohen's kappa coefficient, intraclass correlation coefficient (ICC), or Fleiss' kappa.

: We already conducted the inter-rater and intrarater reliability. We added the results in statistical analysis section.
